# Learning and recognition of tactile temporal sequences by mice and humans

**Michael R Bale[1,2]\*, Malamati Bitzidou[1†], Anna Pitas[1,2†], Leonie S Brebner[1], Lina Khazim[1], Stavros T Anagnou[1], Caitlin D Stevenson[1], Miguel Maravall[1,2]\***

[1]Sussex Neuroscience, School of Life Sciences, University of Sussex, Brighton, United Kingdom; [2]Instituto de Neurociencias, Consejo Superior de Investigaciones Científicas-Universidad Miguel Hernández, Alicante, Spain

**Abstract** The world around us is replete with stimuli that unfold over time. When we hear an auditory stream like music or speech or scan a texture with our fingertip, physical features in the stimulus are concatenated in a particular order. This temporal patterning is critical to interpreting the stimulus. To explore the capacity of mice and humans to learn tactile sequences, we developed a task in which subjects had to recognise a continuous modulated noise sequence delivered to whiskers or fingertips, defined by its temporal patterning over hundreds of milliseconds. GO and NO-GO sequences differed only in that the order of their constituent noise modulation segments was temporally scrambled. Both mice and humans efficiently learned tactile sequences. Mouse sequence recognition depended on detecting transitions in noise amplitude; animals could base their decision on the earliest information available. Humans appeared to use additional cues, including the duration of noise modulation segments.

**\*For correspondence:** m.bale@ sussex.ac.uk (MRB); m.maravall@ sussex.ac.uk (MM)

[†]These authors contributed equally to this work

**Competing interests:** The authors declare that no competing interests exist.

## Introduction

To make sense of the world around us, the brain must integrate sensory patterns and sequences over time and assign them meaning. Signals in our environment unfold over time and can only be interpreted by decoding their temporal patterning. The ability to do so underpins much of our sensory experience – for example, it is central to recognising a favourite melody or a passage of speech (*Wilson et al., 2017*). As first proposed over 60 years ago (*Lashley, 1951*), sequence processing provides a model for investigating how neuronal circuits give rise to object perception and recognition, a central goal of neuroscience (*Griffiths and Warren, 2004*; *Dehaene et al., 2015*).

In tactile sensation, fast sensory events, such as fluctuations in the forces acting on a whisker follicle, are encoded faithfully and with high temporal precision (*Johnson, 2001*; *Johansson and Birznieks, 2004*; *Jones et al., 2004*; *Chagas et al., 2013*; *Bale et al., 2015*; *Campagner et al., 2016*; *Bush et al., 2016*). Exploring an object by scanning with fingertips or whiskers generates a series of tactile events concatenated over time (*Phillips and Johnson, 1985*; *Phillips et al., 1990*; *Weber et al., 2013*; *Maravall and Diamond, 2014*; *Sofroniew and Svoboda, 2015*; *Saal et al., 2016*). Recognising the object as a whole – its texture, shape or size – requires integrating over these events, with the relevant timescales varying from tens of milliseconds (ms) to seconds.

Introspection suggests that meaningful auditory sequences, such as those in speech or music, can be learned quickly and robustly. We wondered whether similarly effective sequence learning and recognition occurs in tactile sensory systems. We further wished to explore the cues that could underlie capacities for tactile sequence recognition.

To address these issues, we developed a new experimental design for testing sequence discrimination in mice and humans. Participants learn to distinguish a target stimulus sequence, constructed from an underlying noise waveform, from other stimuli that differ only in their temporal patterning

over hundreds of milliseconds. Our results demonstrate efficient learning of tactile sequences both in mice and in humans. This behaviour provides an assay for exploring the neuronal circuit mechanisms that underpin recognition of temporally patterned stimuli.

## Results

### Achieving sequence recognition by mice

We sought to train mice to recognise a target stimulation sequence delivered to their whiskers (*Figure 1*). Our aim was for mice to distinguish the target sequence based on the order in which its elements appeared. To this end, mice were trained to distinguish between initially meaningless GO and NO-GO sequences built from series of identical 'syllables', with the sequences differing only in that syllables were scrambled in time over hundreds of milliseconds (each individual syllable lasting 100 ms, for a total of 8 syllables; *Figure 1D*). The initial syllable was identical across GO and NO-GO sequences in order to avoid providing a stimulus onset cue (*Figure 1D*).

Mice (n = 22) were trained to associate the GO stimulus with a water reward by making water available when the GO stimulus was delivered; on the first few days of training no other whisker stimuli were given, so that mice effectively learned to detect whisker stimulation. As soon as animals demonstrated detection (75% correct detection trials), we introduced an initial NO-GO sequence. To make this stage easier, this initial NO-GO sequence consisted of a square wave riding upon low amplitude noise, distinctly different from the GO sequence (*Figure 1D*, 'Square Wave'). Mice quickly learned to distinguish the square wave stimulus from the GO waveform (75% correct; within four sessions; *Figure 2A*). They were immediately moved to the next stage of training to avoid creating an artefactual generalized association between 'noisy' stimuli (as opposed to square waves) and water availability. In the following —more demanding— stage, the NO-GO sequence consisted of a scrambled GO sequence with half (4 of 8) syllables knocked out (*Figure 1D*, 'Half' NO-GO).

Mice accomplished each stage of training within a few days (*Figure 2A*), performing approximately 200–300 trials per daily session (mean 249 trials; SD 71 trials; total n = 456 sessions in 22 mice). During this process, recognition of the GO sequence was mediated by the animal's whiskers: performance fell to chance level upon removing the whiskers from the moving stimulator (*Figure 2B*). Performance recognising the GO sequence was robust against variations in how the sequence was presented: daily changes in the tube's positioning relative to the stimulated whisker did not noticeably affect performance. To test this invariance more specifically, in a subset of experiments mice were trained on a multi-whisker version of the task where whiskers (left untrimmed) were inserted into a wire mesh attached to the piezo actuator. Whiskers were first removed from the stimulator mesh; then, after a period of trials with stimulator movement but no whisker stimulation — during which performance dropped to chance level—, the actuator was rotated 90° and whiskers reinserted into the mesh. Reinsertion and stimulator rotation changed the identity and set-point of whiskers being stimulated as well as the direction of stimulation. Thus, the new configuration involved a different array of forces and moments acting upon a different set of whisker follicles. Yet performance quickly recovered to the level reached before whisker removal (*Figure 2C*; repeated for n = 4 mice; p=0.44; Wilcoxon signed rank test). Sequence recognition therefore transferred across different stimulation configurations, implying that the animal must be focusing on aspects of the pattern of fluctuations over time, rather than on movements of a specific whisker or in a specific direction.

In the final stage of training, the NO-GO sequence comprised identical syllables to the full GO sequence but scrambled in time: that is, syllable ordering changed (*Figure 1D*). An example session from this final stage is shown in *Figure 3*. The animal's performance on this session demonstrates consistent licking to the GO sequence and an ability to withdraw from licking impulsively to the NO-GO sequence. In this session, the rate of lick responses in a GO trial became greater than that in a NO-GO trial by 200 ms into the stimulus sequence (*Figure 3D*). This was 100 ms after the transition from the first to the second syllable, which –given our 100 ms resolution– was the earliest that an instantaneous ideal observer could have determined sequence identity.

Mice that underwent training up to the final stage performed beyond 70% on at least one session (median 73.9%, SD 9.4%; n = 5 out of 6 mice). Animals could maintain their performance over several days, albeit with fluctuations (*Figure 4A*), again despite day-to-day variability in how the whisker

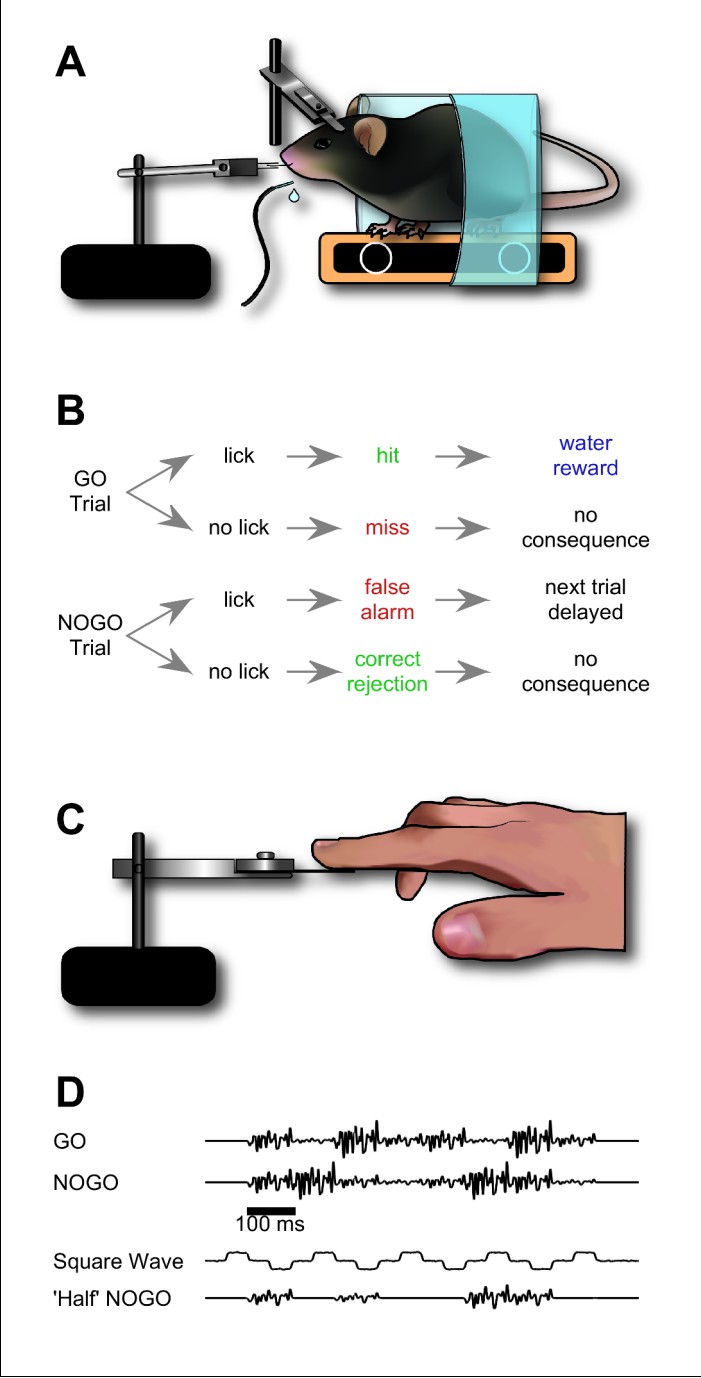

**Figure 1.** Design of sequence recognition task for mice and humans. (**A**) Illustration of the treadmill-based behavioural and stimulus delivery setup for head-fixed mice. (**B**) Block diagram representing the structure of the GO/NO-GO paradigm. On GO trials, a mouse licking the water spout within the response period (hit) was rewarded with a water droplet. If the mouse licked on a NO-GO trial (false alarm), the next trial was delayed by 2–5 s. (**C**) Illustration of the stimulus delivery setup for human experiments. (**D**) Stimulus sequences for GO/NO-GO discrimination and intermediate 'shaping' sequences used for training.

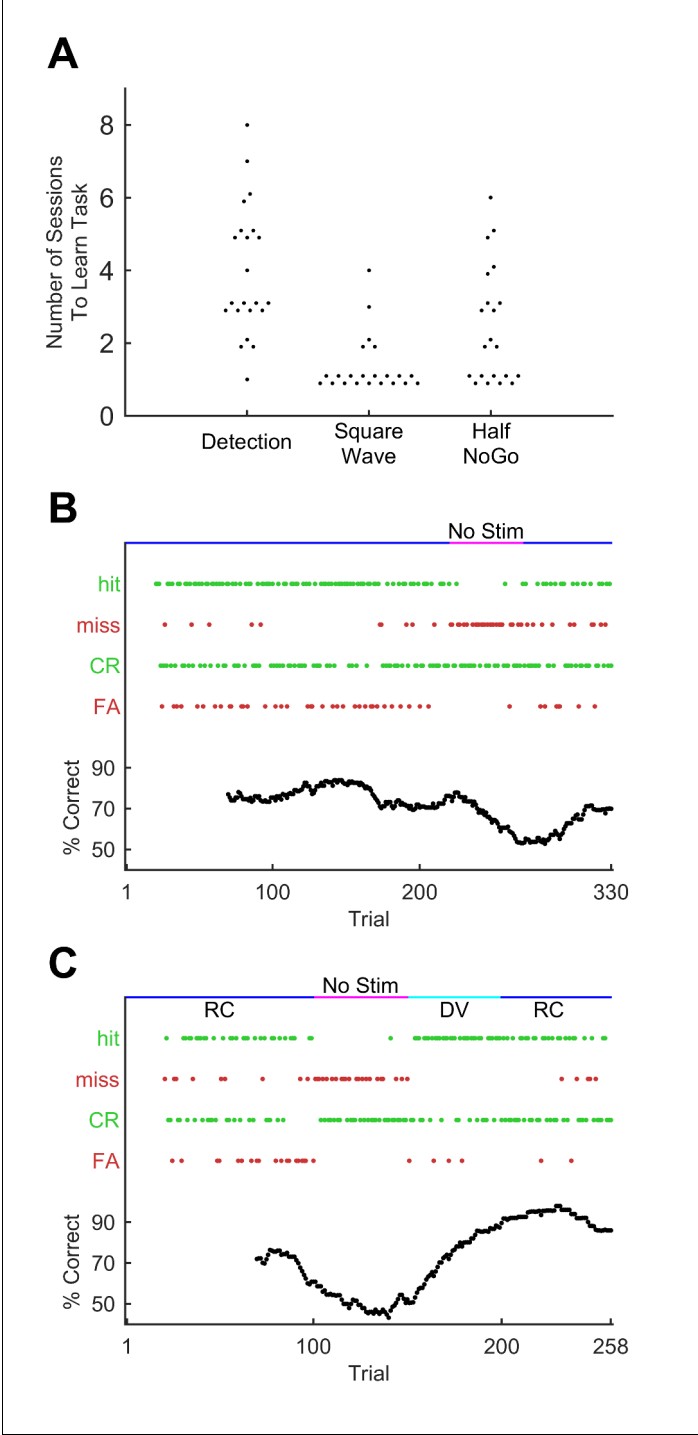

**Figure 2.** Progression through sequence learning in mice. (A) Number of training sessions needed to learn different training stages (75% performance criterion). Each dot, one mouse. Dots are jittered for visualisation. (B) Performance metrics for an example training session (discrimination of the GO sequence from square wave). Correct trials are green dots, incorrect are red dots. Performance quantified as % correct over a 50-trial moving window (bottom). Whisker dependency on the task was verified by removing the actuator at trial 220 (fuchsia bar) and reinserting (blue bar) at trial 270. Apparent delay in performance is caused by 50-trial averaging. (C) Performance for an example session with stimulus rotation (GO sequence versus square wave discrimination). Main symbols as for B. Stimuli were delivered, as normal, first in the rostro-caudal axis (RC; blue bar) but following a brief period of stimulator removal (fuchsia bar), in the dorso-ventral (DV; cyan) axis for 50 trials. Stimulation then returned to RC for the remainder of the session.

*Figure 2 continued on next page*

*Figure 2 continued*

The following source data is available for figure 2:

**Source data 1.** This zip archive includes all files used for generating the figure.

was attached to the stimulator. On every stage of training, improvements in performance occurred mainly through learning to withhold impulsive false alarm responses (licks) (*Figure 3C–D and 4B–C*), in common with other discrimination tasks in mice (*Guo et al., 2014*; *Berditchevskaia et al., 2016*). Thus, high performance was associated with low false alarm rates (*Figure 4C*).

In trained animals, therefore, the rate of licks on GO trials was higher than on NO-GO trials. The time elapsed from trial start to when the rates diverged provides a conservative upper bound on an animal's decision time (*Figure 3D*); we refer to this as 'discriminative lick latency' (see Materials and methods). This latency varied from animal to animal and session to session (*Figure 4D*). For different animals, median discriminative lick latency ranged between 200–550 ms. That this value was at most 550 ms indicates that mice reached their decision on the identity of a sequence well before its end, as sequences lasted 800 ms. Performance on a session did not correlate with early or late detection of the sequence (*Figure 4D*; Spearman rho = - 0.033; p=0.88).

These experiments show that mice learned to distinguish whisker-mediated stimuli that differed from others only in their temporal patterning over hundreds of milliseconds. They further indicate that mice based this behaviour on aspects of the pattern of fluctuations over time; in some cases, mice discriminated effectively almost as soon as the identity of the sequence was detectable.

## Rapid learning of noisy sequences by humans

We tested whether humans could also learn to recognise the same temporally patterned tactile noise stimuli, delivering the patterns through an actuator applied to a fingertip. Participants were asked to indicate recognition of the target sequence by pressing a button. They first underwent a training session in which the GO target sequence —identical to that presented to mice— was inter-leaved with a series of non-target stimuli. Early in the training session, we presented non-target patterns set to be easily distinguishable from the GO pattern (e.g. sinusoidal waveforms). Later, these clearly distinct patterns were replaced by NO-GO sequences that differed from the GO target only in that their constituent syllables were scrambled, as in the final stage of mouse training. Human participants quickly improved their performance over the course of this session, typically converging to a steady level of performance despite the increase in difficulty during the session (*Figure 5A*). This indicated fast learning of the GO sequence. Performance as tested after the end of this learning phase was maintained in a later session separated by at least one week, suggesting remarkable robustness (*Figure 5B*; retest performance was actually higher, although the difference did not reach significance; p=0.0547; n = 11 participants; Wilcoxon signed rank test). Despite unavoidable differences in training procedure, mice and humans reached a similar median level of performance (mice: 74%; humans: 70%), and the range of performance across individuals also overlapped substantially between the two species (*Figure 5B*).

## Potential cues for sequence recognition

Our experiments established that animals based their sequence discrimination behaviour on aspects of the pattern of fluctuations over time. This prompted us to wonder which cues were robust correlates of sequence identity, and which were used under our experimental conditions.

In our task, possible cues ranged across timescales from global to local, as follows (*Figure 6A*). At the highest (most global) level the GO sequence could be recognised by extracting its overall ordering rule (i.e. distinguishing the [3 1 4 2 3 1 4 2] amplitude modulation envelope versus other scrambled orders). However, recognition could also stem from detection of specific syllables within the overall modulation envelope. For example, in the GO sequence [3 1 4 2 3 1 4 2] the second syllable was smaller in magnitude than the first, and this was in contrast to the NO-GO sequence [3 4 2 1 2 4 3 1], whose second syllable was greater than the first (*Figure 1D*). Thus, an animal could potentially base its recognition of the GO sequence on detecting a downwards modulation in noise

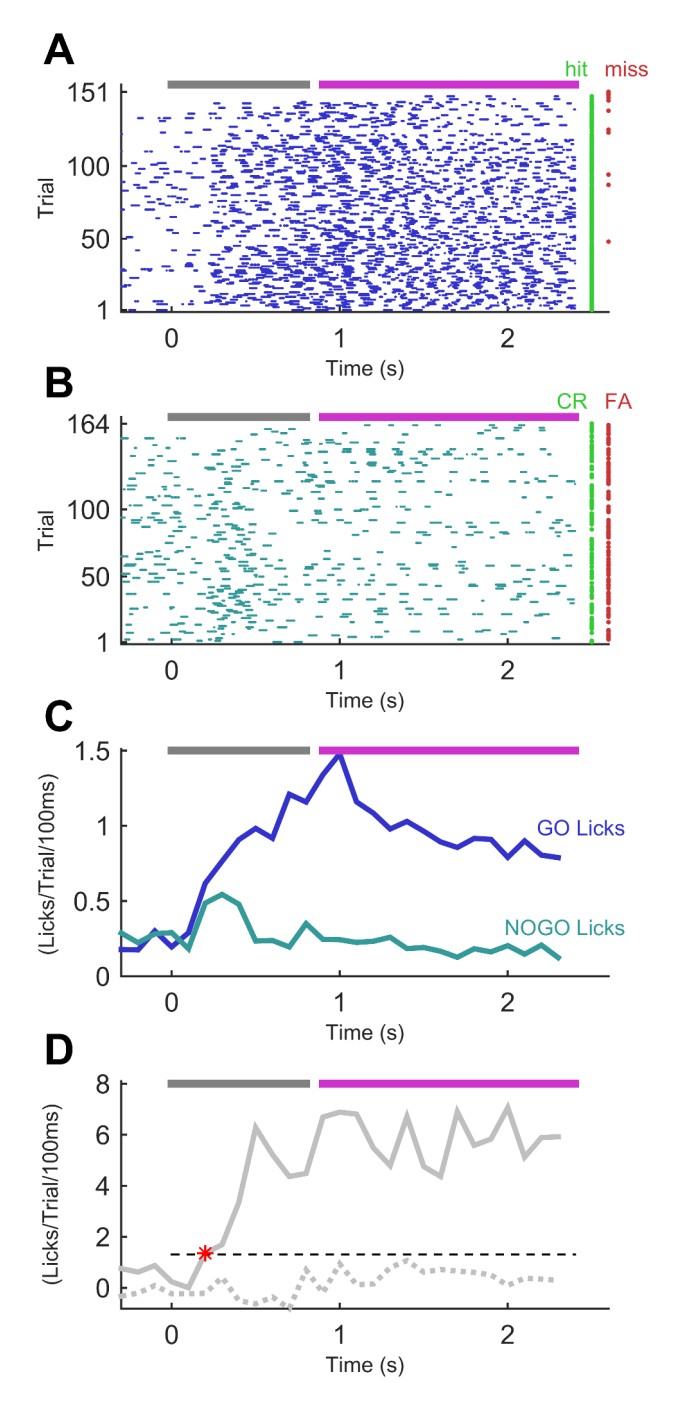

**Figure 3.** Performance of a trained mouse in a sequence recognition session. (**A**) Lick time raster plot for GO trials in an example session on the final stage of training (GO vs 'full' NO-GO). Licks are referenced to the start of the sequence, which lasts 800 ms (grey bar at top). The response period (fuchsia bar) opens after the end of stimulation. Water was only available in the response period. Hit and miss trials (rows) marked with green and red dots, respectively. (**B**) Lick time raster plot for NO-GO trials of same session as A. Green dots denote correct rejection trials, red dots false alarm trials. Other symbols as for A. (**C**) Smoothed average of lick rate signals per trial from A and B. Dark blue, GO trials. Teal, NO-GO. (**D**) Difference of lick rate plots. Continuous thick line is true difference between lick rates on GO and NO-GO trials from the best 50-trial moving window from C. Broken line is average of 50 repeats of differences between randomly drawn trials. Dashed threshold line is 95% confidence interval of lick rate values from the 50 repeats. Asterisk denotes the time when the true lick rate difference

*Figure 3 continued on next page*

*Figure 3 continued*
surpassed the 95% CI, termed discriminative lick latency. This is an upper bound on the animal's time to discriminate sequence identity.
The following source data is available for figure 3:
**Source data 1.** This zip archive includes all files used for generating the figure.

amplitude after 100 ms. Finally, a strategy based on detecting even briefer, sub-syllabic events or 'landmarks' in the sequence, such as fluctuations in whisker velocity happening in a certain relative order (*Figure 6A*), could also be possible. We sought to identify cues and strategies accounting for the performance of mice and humans.

First, our stimulus design ruled out cues based on overall temporal averaging or summing over stimulus parameters throughout the duration of the sequence, because GO and NO-GO sequences consisted of identical, but temporally scrambled elements. Second, mice reached their decision well before the end of sequence presentation (*Figure 4D*): as mentioned, discriminative lick latencies were sometimes consistent with fast detection of the first transitions or syllables that allowed sequence identification. Thus, just one or a few transitions in noise modulation, or the arrival of a small number of stimulus landmarks, could suffice for an animal to reach its decision.

## Binary sequence discrimination

To further test the ability of mice to exploit specific cues for sequence recognition, we designed a version of the task in which the GO target sequence consisted of a simple succession of epochs of large and small noise amplitude: using the same notation as above, [4 1 4 1 1 1 4 1] (*Figure 6B*). A separate set of animals (n = 10) was trained to distinguish this new GO sequence from a NO-GO sequence that, as before, differed only in its temporal patterning: [4 1 1 1 4 1 4 1] (*Figure 6B*). These sequences were simpler than the original design in that they were binary: their constituent syllables were only 'large' (4) or 'small' (1). Note that the pattern design was such that the actual sequence of transitions in amplitude was identical between GO and NO-GO: transitions followed the order [−3 +3 −3 +3 −3]. Thus to distinguish these sequences, animals needed to focus on syllable ordering or on the different durations of small-amplitude epochs: for example, in the binary GO target sequence the first 'small' epoch lasted just 100 ms, but in the NO-GO it lasted for a total of 300 ms (because comprising three syllables).

We found that mice did not perform well at distinguishing [4 1 4 1 1 1 4 1] from [4 1 1 1 4 1 4 1], despite having been trained exclusively on this variant of the task. Performance was above chance but never reached 75% (*Figure 6C*; n = 3), in contrast to the original variant. In particular, animals consistently displayed high false alarm rates, suggesting that they failed to detect what made the binary NO-GO stimulus different from the binary GO (data not shown). This suggested that in this paradigm mice could have had trouble either detecting the simpler, binary stimulus modulation epochs or recognising their differential duration.

To distinguish between these possibilities, we tested performance on probe sessions with two variants of the binary NO-GO sequence. In the first, the binary GO target remained identical as [4 1 4 1 1 1 4 1], but the NO-GO sequence was [4 1 1 1 1 1 4 4]. This alternative NO-GO stimulus had the same number of large and small syllables as the one in the previous paragraph, but a different temporal arrangement, with all small-amplitude syllables appearing consecutively and forming a single very long central period (*Figure 6B*). The sequence therefore effectively had just two 'large' epochs, at the beginning and at the end, and a single very long 'small' epoch. There were now just two transitions in amplitude, in the order [−3 +3]. We therefore expected this NO-GO sequence to be easier to discriminate from the GO sequence despite having the same overall energy. In the second variant, the binary GO target also remained identical and the NO-GO stimulus was [4 1 1 1 1 1 1 1] (*Figure 6B*). The goal of this variant was to check whether mice could straightforwardly distinguish large or small syllables, as the NO-GO stimulus consisted mostly of small syllables and had lower overall energy than the (unchanged) GO stimulus.

In both of these variants animals performed better than in the original binary design (*Figure 6C*; $p=1.8\times10^{-10}$; t-statistic for the regression on task type = 6.82; n = 10 mice and n = 161 sessions;

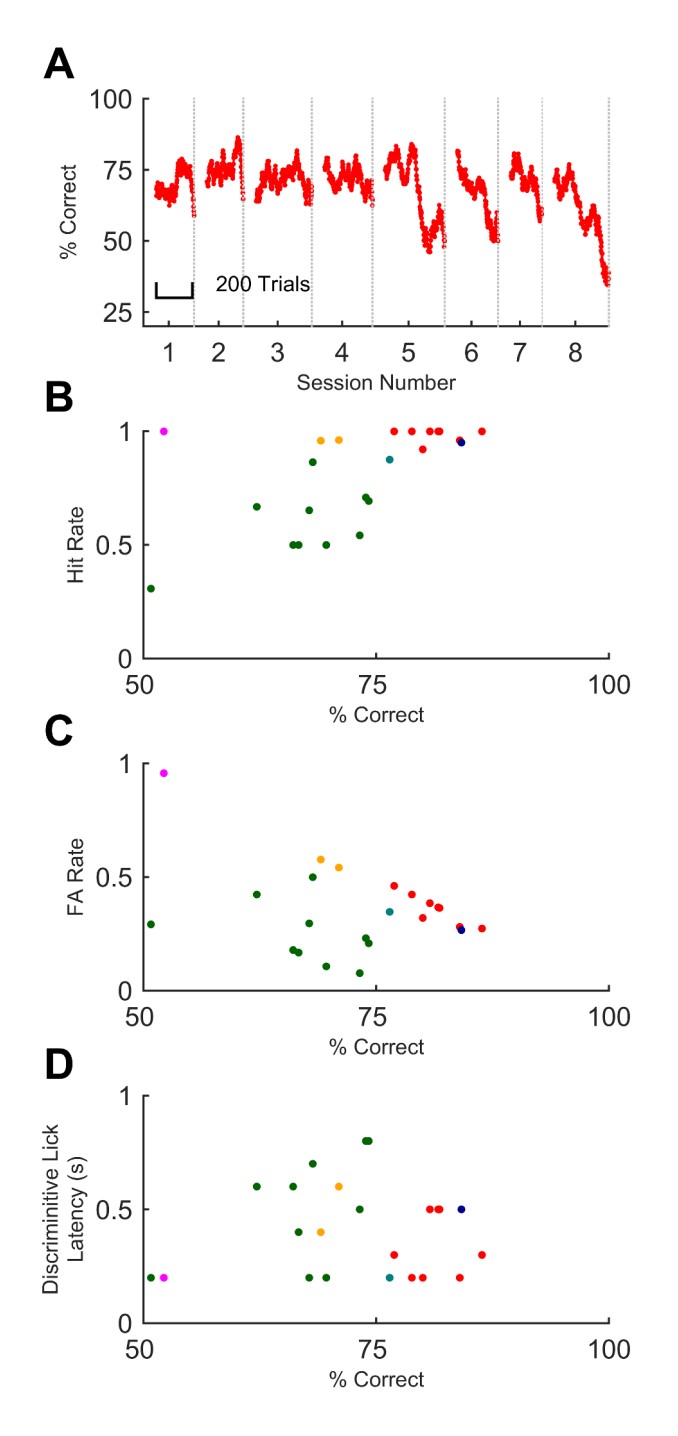

**Figure 4.** Sequence recognition performance across mice. (**A**) Performance on full sequence recognition task across 8 successive behavioural sessions for one mouse. Performance is averaged over a 50-trial moving window. Colour corresponds to colour of same mouse in panels B-D. (**B**) Hit rate plotted against % correct for all sessions on full task. Each dot, one session on GO vs full NO-GO; each colour depicts a different mouse (n = 6). (**C**) False alarm rate plotted against % correct. Symbols as for B. (**D**) Discriminative lick latency plotted against % correct.

The following source data is available for figure 4:

**Source data 1.** This zip archive includes all files used for generating the figure.

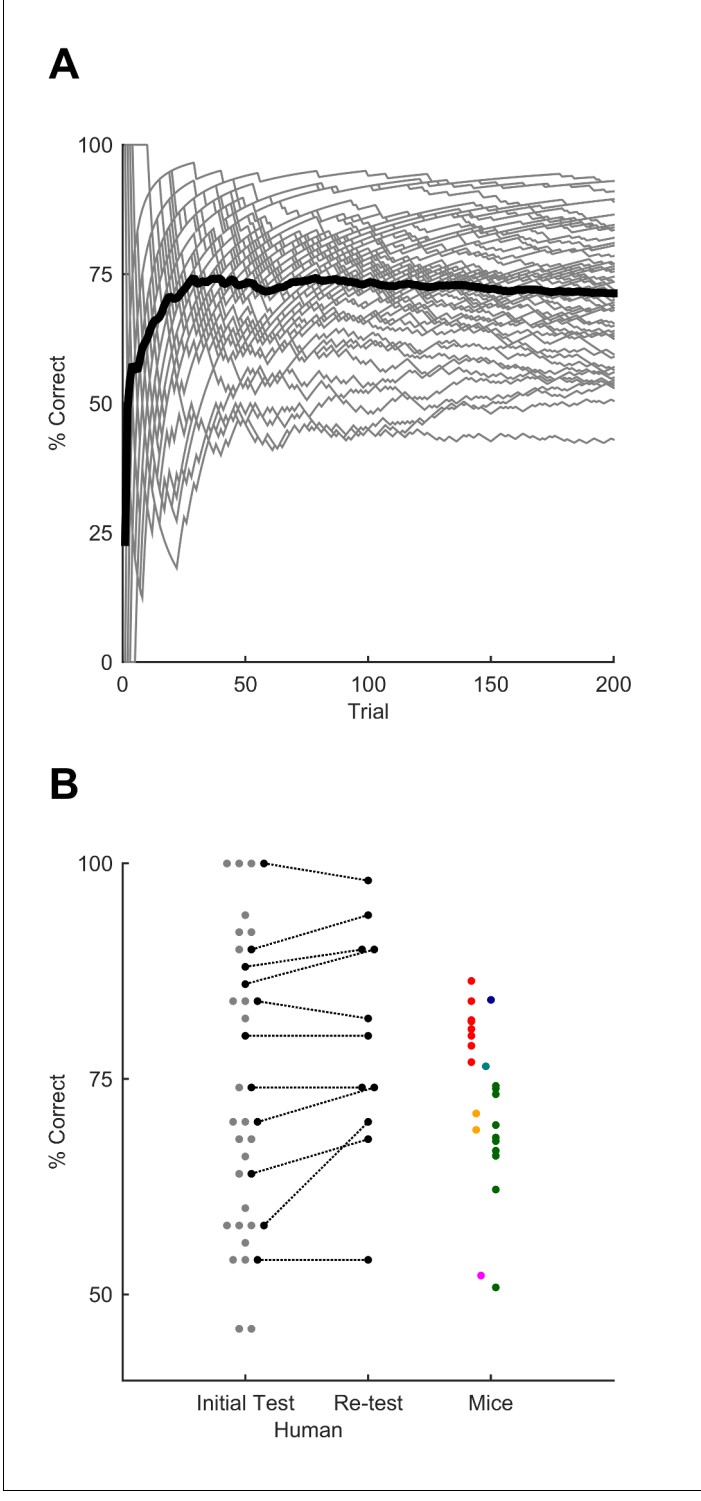

**Figure 5.** Sequence recognition performance across humans. (**A**) Performance over the course of the first training session, measured as % correct averaged over 50-trial moving window. Grey lines, individual participants. Black line, average over participants. (**B**) Performance for best 50-trial window on initial test session after training and (on a subset of participants) upon retesting after a minimum of one week. Each grey dot, one human participant; black dots, participants tested on both sessions. Coloured dots, performance on individual sessions for mice, replotted from *Figure 4* for comparison (each mouse, one colour; colours as for *Figure 4*). Dots are jittered along x axis for visualisation.

*Figure 5 continued*

The following source data is available for figure 5:

**Source data 1.** This zip archive includes all files used for generating the figure.

generalised linear mixed-effects model). Note that performance in the simpler variants was at a level comparable to that of the original GO vs NO-GO paradigm (*Figure 6C*). Mice also successfully distinguished the binary GO sequence from noise stimuli with no modulation, i.e. with constant noise amplitude: [1 1 1 1 1 1 1 1] or [2 2 2 2 2 2 2 2] (percentage correct >75% for all animals for [1 1 1 1 1 1 1], 3 out of 4 for [2 2 2 2 2 2 2 2]; n = 4 mice; data not shown).

These results indicate that mice in the binary sequence experiments could successfully detect 'large' epochs and recognise their number, and could use relative modulations (i.e. transitions) in noise amplitude as cues. However, mice trained in these experiments were not as successful at using the duration of modulation epochs as a cue.

In contrast to mice, humans rapidly performed well on the [4 1 4 1 1 1 4 1] vs [4 1 1 1 4 1 4 1] version of the task (*Figure 6C*), even when they had had no prior exposure to the version in *Figure 5* (5 out of 8 participants). They achieved high performance within one training session. They also subjectively reported being able to use the relative durations of 'small' and 'large' epochs as a cue. Thus, under these conditions humans appeared to readily access more cues for sequence discrimination than mice, including the duration and ordering of intervals in noise modulation.

## Fixed landmarks versus amplitude modulation

Our findings suggested that humans could discriminate sequences based on multiple cues. We wondered whether, in addition to being sensitive to the size and timing of noise amplitude modulations, human participants might also rely on detecting learned sub-syllabic 'landmarks', that is, specific brief events or fluctuations within the stimulus waveform (*Figure 6A*). To address this, we assessed whether the presence of fixed waveforms influenced performance.

Upon training participants on the initial version of the task (*Figure 5*), we tested performance on a variant of the design with two types of GO trials. The first type of trial used a target sequence constructed by applying amplitude modulation to a waveform that was identical (repeated) across syllables and trials ('frozen'). This sequence was used throughout training and in the experiments of *Figures 1–5*. The second type of trial presented a sequence built by modulating a noise waveform that varied on every repeat ('unfrozen') (*Figure 6D*). For unfrozen sequences, each of the eight syllables was based on a different noise snippet and each trial was constructed from a fresh waveform. Thus, in this type of trial, the amplitude modulation envelope characteristic of the GO sequence remained identical across target trials, but not the precise stimulus values, so that sub-syllabic fluctuations were not conserved. Frozen and unfrozen GO trials were interleaved within a session. Note that unfrozen waveforms could vary in their empirical standard deviation, potentially leading to a confound caused by variability in perceived stimulus amplitude. To control for this, we included in our analysis only stimuli matched for empirical standard deviation. We compared hit rates for both types of GO trial (*Figure 6E*). Hit rates varied little across type of trial (frozen trials mean 0.76, SD 0.17; unfrozen trials mean 0.73, SD 0.19; p=0.09; n = 27 participants; Wilcoxon signed rank test). Thus, participants did not require specific brief waveform landmarks to achieve sequence recognition.

In conclusion, humans could use cues based on ordering, timing and feature detection to recognise a target tactile temporal sequence. Mice tested with an identical stimulus paradigm achieved similar recognition of a sequence delivered to their whiskers, but appeared to base their performance primarily on detecting the presence of particular relative changes (transitions) in noise amplitude.

## Discussion

Senses such as touch or hearing depend critically on the detection of temporal patterning over timescales from tens of milliseconds to seconds: in these sensory modalities, signals unfold over time and

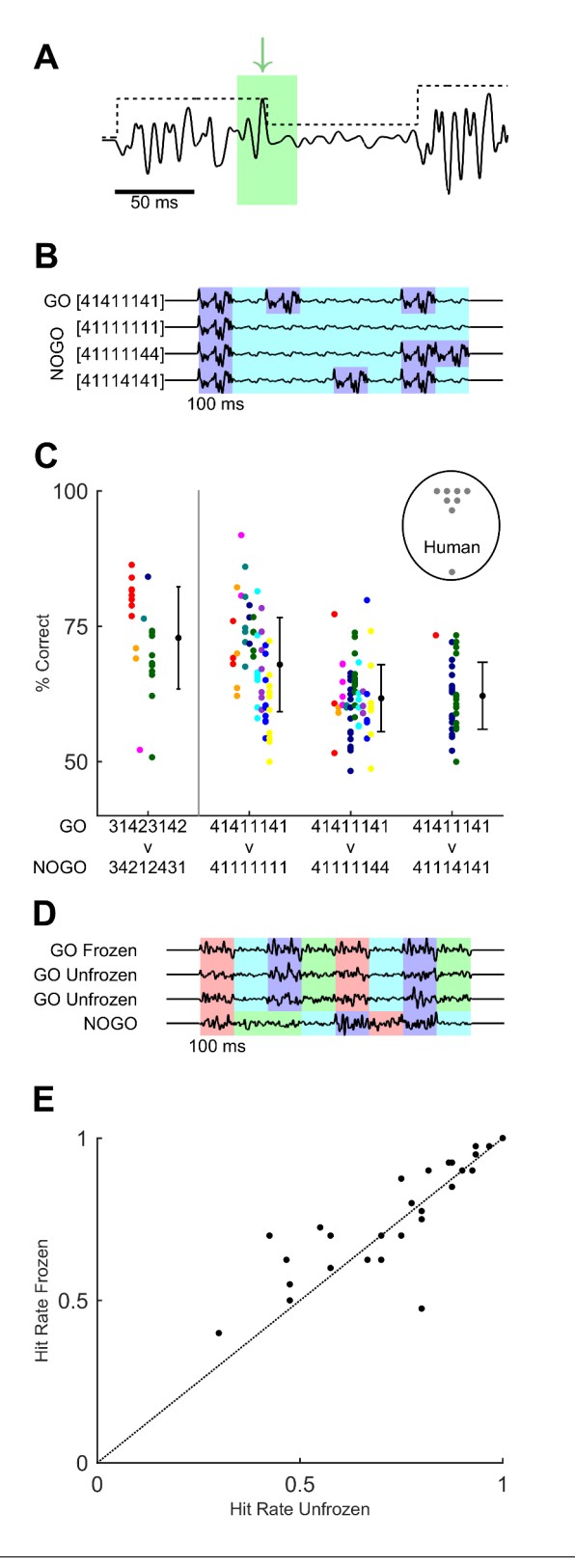

**Figure 6.** Variants of task design to test for behavioural use of cues. (**A**) Cues within the GO sequence (black line) that could allow recognition of sequence identity. An example of a local cue (within green box) is the large isolated transient 'landmark' (green arrow) immediately followed by a low amplitude syllable. In contrast, global cues involve changes in integrated stimulus amplitude over time, as reflected in the amplitude modulation

*Figure 6 continued on next page*

*Figure 6 continued*

envelope (black dotted line). (**B**) Binary sequences distinguishable based on syllable ordering or the durations of small-amplitude epochs. Different colours indicate epochs of large and small noise amplitude. (**C**) Performance on task variants using binary sequences, compared to original GO vs full NO-GO design (leftmost data point, replotted from *Figure 4 and 5* with same colour assignment). Each colour depicts a mouse. Black dots and error bars, grand average and SD for each task. Grey dots in labelled circle depict performance of individual human participants on [4 1 4 1 1 1 4 1] vs [4 1 1 1 4 1 4 1]. (**D**) Sequences used to test effect of fixed 'landmarks'. Frozen GO used an identical waveform across syllables, trials and sessions. Unfrozen GO maintained the same sequence of noise amplitudes (indicated by colour coding) but varied the detailed waveform across syllables, trials and sessions. NO-GO scrambled the order of syllables, i.e. the sequence of noise amplitudes. (**E**) Hit rate of humans on frozen and unfrozen GO trials. Each dot, one participant and session.

The following source data is available for figure 6:

**Source data 1.** This zip archive includes all files used for generating the figure.

are incomprehensible if the temporal relationship between their elements is lost. Sequence learning and the processing of temporal duration are impaired in psychiatric disorders including depression and schizophrenia (e.g. [*Fischer, 1929*; *Marvel et al., 2005*; *Siegert et al., 2008*; *Abrahamse et al., 2009*]). Here, we developed an assay suitable for evaluating tactile sequence discrimination. Mice learned to distinguish a target stimulation sequence delivered to their whiskers: the sequence differed from others only in its temporal ordering over hundreds of milliseconds. Humans receiving identical sequential stimuli applied to their fingertip also rapidly learned to perform the task. There was substantial overlap between the performance levels of mice and humans (*Figure 5B*).

Similar human paradigms have been used to discover implicit learning of meaningless auditory noise patterns (*Agus et al., 2010*), demonstrating that patterned noise learning generalises across species and sensory systems. In rodents, our design provides an assay for elucidating how neurons within sensory circuits respond and interact under temporally patterned stimulation.

Mechanisms for memorising and determining sequence identity regardless of syntax and semantics have been proposed to be precursors to speech recognition (*Petkov and Jarvis, 2012*; *Comins and Gentner, 2014*; *Wilson et al., 2017*). In our paradigm, sequences were built from chunks of noise with no semantic content or prior meaning. Structural rules such as those aiding interpretation of music or speech (grammar, syntax) were not present as cues. The protocol presented here involved learning only a single instance of a GO target sequence, and did not test generalisation and rule abstraction. However, it is straightforward to modify the present design to one whereby decisions must be based on progressively more general sequencing or branching rules. For example, mice robustly learn a simpler version of the task where the GO sequence is structured as XABX and the NO-GO sequence as XBAX (with X, B and A denoting different waveforms that, concatenated, make up the sequence; Bale, Bitzidou, Giusto and Maravall, data not shown). These stimuli can be lengthened as needed in order to vary sequence complexity: e.g. the GO target could become XABCX and NO-GO stimuli could be XBACX (distinguishable from the target at the transition from X to B) or XACBX (distinguishable from the target only later, at the transition from A to C). Mice also learn to distinguish XABX versus XBAX in an auditory two-alternative forced choice design for freely moving animals (Saska, Giusto, Bale, Bitzidou and Maravall, data not shown), suggesting that this form of sequence learning generalises across sensory modality and protocol.

In both rodents and humans, tactile sensation commonly involves active exploration, whereby the animal generates sensor motion in order to feel the object. Our design departed from active sensing in that mice and humans were trained to receive and recognise sequences delivered by a stimulator. Sequential transitions in texture may be encountered by mice running along walls or tunnels (*Jenks et al., 2010*; *Sofroniew et al., 2014*), and fingertips are sensitive to changes in pressure received passively; 'receptive' sensation is also routinely mediated by whiskers (*Diamond and Arabzadeh, 2013*). Rather than fully mirroring a situation faced either by a mouse or a human in real life, our experimental paradigm sought to isolate behavioural capacities for learning and recognising temporal patterning, which is a key aspect of the structure of sensory environments. Our choice of design specifically allowed us to test for learning of arbitrarily patterned stimuli, which both humans

and mice were able to accomplish. Other laboratory-based sensory discrimination tasks similarly seek to abstract important attributes of stimuli in the spatial or temporal domain, rather than fully reproducing a natural situation (*Romo and de Lafuente, 2013*). A classical approach in comparative cognition employs tasks that are not part of an animal's natural repertoire to challenge its capacities (*Gould, 2004*; *Roth and Dicke, 2005*; *Alem et al., 2016*; *Ishiyama and Brecht, 2016*; *Loukola et al., 2017*).

Although mice and humans were both able to learn to recognise a sequence identifiable by the order of its elements, in our design humans accessed a wider range of cues than mice. It is difficult to separate this result from the obvious differences in our ability to communicate task parameters to humans and mice. Further, we cannot rule out that mice have access to the same cues as humans but that only some cues were engaged by our design. With these caveats, our results suggest that mice relied primarily on particular transitions in stimulus amplitude (and therefore in kinetic or kinematic attributes). Mice could detect 'large amplitude' epochs and recognise their number, and use transitions in noise amplitude as cues. In the experiment shown in *Figures 2–4*, mice licked differentially for the GO target sequence well before its end (*Figure 3D and 4D*), suggesting that they identified the target by detecting particular changes in noise amplitude occurring early in the sequence. In the experiment in *Figure 6B–C*, mice had trouble distinguishing the target sequence from others with the same number of 'large' and 'small' epochs and transitions. They did successfully recognise the GO sequence compared to stimuli where the noise amplitude remained constant, regardless of whether the integrated energy of those stimuli matched or exceeded that of the GO sequence, and with no need for prior training (data not shown). This implies that animals used sensitivity to relative changes in noise amplitude as a behavioural cue, a capacity previously demonstrated in rats under a more cognitively demanding task design (*Fassihi et al., 2014*). Further testing of mouse capacities for using additional cues, from epoch duration up to more abstract sequencing rules, is needed.

Humans appeared to use global and local information on fluctuations in stimulus amplitude to arrive at a heuristic for sequence recognition. Participants receiving the GO sequence often reported feeling a vibration consisting of a rhythmic series of buzzes, or counting 'beats' in the stimulus. This denoted an ability to detect stimulus periodicity or repeats. However, no participants reported overt awareness of a change in the order of explicitly recognised sequence elements. In a previous auditory study, human listeners were asked to report when a noise stimulus consisted of concatenated repeats of an identical 500 ms segment as opposed to a single 1 s segment (*Agus et al., 2010*). Listeners improved their ability to detect the repeated-noise stimuli when they were unwittingly exposed to the stimulus a few times, and this improvement in performance seemed related to the learning and detection of low-level stimulus waveform features (i.e. particular structures appearing in the noise) (*Agus et al., 2010*; *Andrillon et al., 2015*). The presence of certain learned features, appearing with a specific temporal relationship to each other, may provide an elementary cue for discriminating and recognising sequences on timescales of hundreds of milliseconds to seconds across modalities. Determining the duration of relevant features and how they are encoded (*Jadhav et al., 2009*; *Waiblinger et al., 2015a*, *2015b*) is a further important task.

Future work must examine how neuronal circuits detect and recognise temporally patterned stimulation sequences. Neurons in early stages of sensory pathways transform any temporally patterned sensory signal into a sequence of precisely timed spikes, so recognising a sensory stimulus with a characteristic temporal pattern –e.g., to discriminate one tactile texture from another (*Weber et al., 2013*)– ultimately implies a need for circuits in higher brain areas to decode a spatiotemporal spike sequence. For the paradigm explored here, this capacity is likely to reside within the neocortex, as suggested by the following findings. In the rodent whisker system, neurons in subcortical stages and primary somatosensory cortex display limited temporal integration (*Maravall et al., 2007*; *Jacob et al., 2008*; *Petersen et al., 2008*; *Stüttgen and Schwarz, 2010*; *Estebanez et al., 2012*; *McGuire et al., 2016*; *Pitas et al., 2017*). Therefore, integration over time to represent specific whisker stimulation sequences must be carried out by higher cortical circuits (*Lim et al., 2016*; *McGuire et al., 2016*; *Fassihi et al., 2017*; *Pitas et al., 2017*). That mice were able to generalise the task across different whisker stimulation directions (*Figure 2C*), which would have evoked responses in different subsets of neurons at earlier stages in the pathway (*Bale and Petersen, 2009*), suggests further evidence for higher cortical task involvement. A hierarchical scheme whereby later stages of cortical processing can integrate stimuli over longer timescales is consistent with findings in primates (*Hasson et al., 2008*; *Honey et al., 2012*).

Which mechanisms contribute to setting integration timescales? Single neurons can be sensitive to spatiotemporal input sequences (*Segundo et al., 1963*; *Branco et al., 2010*; *Baker et al., 2016*). Learning to detect a specific sequence (*Hardy and Buonomano, 2016*) can be accomplished by spike timing-dependent plasticity (*Masquelier et al., 2008*; *Klampfl and Maass, 2013*; *Gütig, 2014*). Timescales for integration of sequences could be regulated by activation of local inhibition (*Kepecs and Fishell, 2014*). Sequence-selective responses can emerge as a result of sensory exposure to the target (*Gavornik and Bear, 2014*). Finally, heterogeneous timescales for integration across cortical processing stages may arise from differences in large-scale connectivity across areas (*Chaudhuri et al., 2014*, *Chaudhuri et al., 2015*). It remains to be determined how these and other mechanisms come together to implement sequence recognition in cortical circuits in vivo.

## Materials and methods

### Surgical procedures

All procedures were carried out in accordance with institutional, national (Spain and United Kingdom) and international (European Union directive 2010/63/EU) regulations for the care and use of animals in research. Details of head bar implantation surgery have been described elsewhere (*Guo et al., 2014*; *Bale et al., 2015*). Briefly, under aseptic conditions, mice (male, C57BL/6J, RRID: IMSR_JAX:000664, total n = 32, 6–9 week old) were anaesthetised using 1.5–2.5% isoflurane in $O_2$ and placed into a stereotaxic apparatus (Narishige, Japan) with ear bars previously coated with EMLA cream. We monitored anaesthetic depth by checking spinal reflexes and breathing rates. Body temperature was maintained at 37°C using a homeothermic heating pad. Eyes were treated with ophthalmic gel (Viscotears Liquid Gel, Novartis, Switzerland) and the entire scalp was washed with povidone-iodine solution. An area of skin was removed (an oval of 15 mm x 10 mm in the sagittal plane) such that all skull landmarks were visible and sufficient skull was accessible to securely fix a titanium or stainless steel head bar. The exposed periosteum was removed and the bone was washed using saline solution. The bone was dried and then scraped using a scalpel blade to aid bonding of glue. Cyanoacrylic glue (Vetbond, 3M, USA) was applied to bind skin edges to the skull and as a thin layer across the exposed skull to aid bonding to the dental acrylic. A custom titanium or stainless steel head bar (dimensions 22.3 × 3.2×1.3 mm; design by Karel Svoboda, Janelia Farm Research Campus, Howard Hughes Medical Institute) (*Guo et al., 2014*) was placed directly onto the wet glue centred just posterior to lambda. Once dry, we fixed the head bar firmly in place by applying dental acrylic (Lang Dental, USA) to the head bar (on top and behind) and the skull (anterior). Mice were given buprenorphine (0.5 mg/kg, I.P.) and further EMLA cream to the paws and ears. Once the acrylic was set, anaesthesia was turned off. Animals were housed individually on a reverse 50:50 light-dark (LD) cycle and allowed to recover for one week post-surgery.

### Head fixation and water delivery

Mice were trained using a shaping procedure to freely enter a head fixation device (*Figure 1A*). We used two device designs. One design consisted of an acrylic tube (32 mm internal diameter) with its head end cut to enable access to implanted head bars. The tube was placed on Parafilm or a rubber glove and clamped into a v-shape groove. This support acted to stabilise the tube, collect faeces and prevent mice from grasping stimulus apparatus and the lickport. The second design consisted of a platform with a custom-made treadmill on which mice could locomote freely (design by Leopoldo Petreanu, Champalimaud Centre for the Unknown). A mesh was fixed over the treadmill to surround the mouse's body, allowing the animal to feel comfortably enclosed rather than exposed. The ends of the head bars were inserted into grooves on two head fixation clamps and tightened using thumbscrews. The head fixation set-up was adapted from (*O'Connor et al., 2010*; *Guo et al., 2014*).

Water was available to mice via a spout made from a blunted gauge 13 syringe needle. Water delivery was controlled via a solenoid valve (LDHA1233215H, The Lee Company, France). The acrylic tube or head bar holder was lined with aluminium foil. Terminals from an A/D input of a signal processor (RP2.1, TDT, USA) were then connected to the water spout and the foil. Tongue contacts with the lick port created brief elevations in voltage consistent with lick durations (*Hayar et al., 2006*).

## Water restriction

To motivate mice to learn and perform the task we employed a water restriction protocol (*Guo et al., 2014*) and made water available as a reward during the task. Mice cope better with water control than food control (*Tucci et al., 2006*). Unless rodents are motivated by fluid or food control, they can fail to learn even simple sensory tasks (*Carandini and Churchland, 2013*) and perform too few daily trials for data collection to be satisfactory. We verified that mice were not motivated by sugary treats alone (Lucozade and chocolate milk). We observed a mild increase in motivation when mice were given sunflower seeds before tasks.

Mouse water intake was regulated so that animals were motivated to perform at around 75% success rate for 200 or more trials per session under our conditions (45–55% humidity, 23°C and atmospheric pressure; reverse 50:50 LD cycle), while remaining active and healthy. This was achieved with two different schedules, depending on the institution where the experiment took place. In one schedule (Instituto de Neurociencias), we titrated down water availability to the amount required for mice to maintain >75% of initial body mass in the short term and gradually increase body mass in the long term (0.5 ml daily including experimental water rewards collected during the session, 7 days a week). In the other schedule (University of Sussex), mice were restricted to 50% of their average free water intake but given free access to water for a finite period during the dark phase of their LD cycle. Body weight (mass) was monitored throughout the study, and we measured experimental reward water intake by weighing mice before and after the daily behaviour session together with collected faeces. For both schedules, mice initially lost weight but then gradually increased body mass over the course of the experiment. Sensory discrimination training began after 9 days on water control.

## Animal handling and training

We initiated water control one week after head bar implantation, and began to handle animals daily. On days 1 and 2 animals were introduced to the experimenter. On days 3 and 4 animals were introduced to the head fixation device. On days 5 and 6 mice received water via a syringe only when inside the device (but not head-fixed). On days 7 and 8 animals were given a sunflower seed and after ingestion were head-fixed and given water via a syringe. Animals became accustomed to head fixation and expected to receive water from the spout situated in front of their head. On day 9, under light isoflurane anaesthesia (1–2%) all whiskers apart from C2 were trimmed bilaterally. At least 30 min later mice began the task. Mice performed a single daily training session. Animals were trained in the dark; illumination, if necessary, was provided by a red lamp.

## Stimulus delivery and design

Our aim was to develop a task whereby tactile sequences delivered to the animal could only be distinguished by discriminating their temporal patterning. Careful control of stimulation patterns was therefore required. To achieve this we delivered controlled stimuli, which animals needed to sense by operating in a 'receptive' mode rather than by active whisking (*Diamond and Arabzadeh, 2013*). In this design, whiskers were inserted into a small tube. Stimulus sequences were generated as filtered noise vibrations, such that whisker stimulation was continuous during a trial (*Figure 1D*). We thus avoided temporally isolated discrete movements that could have initiated whisking or confused the animal as to the start, content and ending of the temporal pattern. Upon head fixation at the start of a session, the left C2 whisker was inserted into a snugly fitting tube (pulled 1 ml plastic syringe) glued to a piezoelectric actuator wafer (PL127.11, Physik Instrumente, Germany). The wafer was mounted vertically and motion was rostrocaudal. In some experiments, a different method to deliver stimuli was required: a metallic 10 mm$^2$ mesh grid was glued to the end of an actuator to enable multiple whisker stimulation and allow quick transition between experiments, as in *Figure 2B,C*.

Stimulus sequences were constructed in Matlab (Mathworks, USA; provided as *Source code 1*), and played via a signal processor (RP2.1, TDT, USA) controlled with code custom-written in ActiveX software (TDT; provided as *Source code 1*). The GO sequence lasted 800 ms and consisted of 8 consecutive 'syllables', where each syllable was a 100 ms segment constructed from white noise with one of 4 amplitude levels (*Figure 1D*). We constructed the sequence as follows: (1) we created a 100 ms white noise snippet generated at a sampling rate of 12207 Hz (in Matlab), (2) stitched 8 snippets

together, (3) multiplied the resulting chain of repeated white noise snippets by an amplitude modulation envelope, (4) convolved this sequence with a Gaussian waveform (SD 1.64 ms) to implement frequency filtering, and (5) normalised the sequence to match the dynamic range of the piezoelectric actuator. In the resulting GO sequence, constituent syllables differed in amplitude: the pattern of noise amplitude modulation was [3 1 4 2 3 1 4 2], with 1 being the lowest amplitude level and 4 the highest. The NO-GO sequence in the full version of the task contained the exact same syllables but in a scrambled order (*Figure 1D*), specifically [3 4 2 1 2 4 3 1]. The target and non-target sequences were therefore identical for the initial 100 ms. Further sequences were created to aid learning and to explore the nature of recognition, as detailed in Results.

We considered other approaches to sequence design before opting for continuous noise stimulation. For example, a sequence can be constructed as a series of discrete whisker pulse stimuli separated by pseudorandom intervals (*Pitas et al., 2017*). We preferred continuous stimulation for two reasons. First, in a sequence consisting of an extended series of 'silent' intervals, an animal can confuse such an interval with the end of the sequence, unless a supplementary cue is provided to signal that stimulation is ongoing. In continuous stimulation, this possible confound is absent, as stimulation remains on throughout the duration of the sequence. Second, any active whisks generated by the animal could interfere with its judgment of the duration of intervals. This potential conflict, inherent to the use of intact whiskers as active sensors, is reduced in the case of continuous stimulation where the animal is being asked to judge the amplitude of 'syllables' lasting 100 ms.

## Task control and analysis

We trained mice to respond to the GO sequence by licking a spout to receive a water reward (1–2 µl). On presentation of the NO-GO sequence mice were trained not to lick (*Figure 1B*). The trial began with the 'stimulation period' (0.8 s) where the sequence was delivered to the whisker. At the end of the stimulation period followed a 'response period' (1.5 s) where mice must lick or refrain depending on the stimulus sequence. Following the GO sequence, if mice licked during the response period (a hit trial) they received a water reward; if they failed to lick (a miss trial) the next trial began as normal. Following a NO-GO sequence, if mice correctly withheld licking during the response period (a correct rejection trial) the next trial began as normal; if they licked (a false alarm trial) the next trial was delayed by 2–5 s. Trial parameters were defined in Matlab using a custom made GUI and then loaded to the RP2.1 signal processor (all provided as *Source code 1*). Trial outcomes were recorded in Matlab using custom-written code (provided as *Source code 1*).

Several related measures can be used to quantify performance, including overall percentage of correct trials, hit rate and false positive rate, and d' (*Green and Swets, 1966*; *Carandini and Churchland, 2013*). Here we present results mostly as percentage of correct trials measured over a 50-trial sliding window during the course of a session. To calibrate this performance measure in terms of statistical significance level, we shuffled stimulus identity and behavioural response (lick/no lick) on a trial by trial basis for each individual session in a test data set of 104 sessions (n = 7 animals; shuffling repeated 10000 times per session). Performing shuffling separately for each session allowed us to control for variations in overall lick rate from animal to animal and during the course of training. For all sessions in the test data set, the probability of achieving 75% correct performance given a random relationship between stimulus and responses was lower than p=0.001; the probability of achieving 70% correct given such a relationship was under p=0.015.

During training, we routinely varied the proportion of GO and NO-GO trials during a session in order to aid learning and keep animals motivated (e.g. the fraction of GO trials could temporarily increase). This could lead to a misleading value of the performance measure. For example, consider a randomly performing mouse that licked on 90% of trials. In a hypothetical 50 trial period with 40 GO and 10 NO-GO trials, it would reach a 90% hit rate on GO trials and a 90% false alarm rate on NO-GO trials. Overall performance would then be 74% correct (=0.8×90%+0.2×10%), despite the mouse performing at chance with no differentiation between GO and NO-GO stimuli. To correct for this, we rebalanced the percentage correct measure so that GO and NO-GO trials are set to have equal weight. This rebalanced measure reports the above hypothetical example as 50% correct (=0.5 × 90% + 0.5 × 10%).

We computed how the rate of licks evolved over time during a trial. Animals that performed well on the task licked more on GO than NO-GO trials (*Figure 3*). The time at which the lick rates for both types of trial began to diverge is an upper bound on an animal's decision time, because the

mouse must have made the decision by the time it displays its response. To determine this 'discriminative lick latency' we first subtracted the lick rate curve for NO-GO trials from that for GO trials (*Figure 3D*). We defined a threshold for when this subtracted curve became positive (i.e. when GO licks surpassed NO-GO licks) as follows. We constructed 50 fake sets of GO and NO-GO trials by picking trials at random from the overall data set. For each of the 50 repeats we determined the corresponding subtracted lick rate curve. We then determined the 95% confidence limit for the resulting distribution of lick rate throughout the trial (time points sampled at 100 ms resolution). We defined discriminative lick latency as the time when the true subtracted curve for hit GO and correct rejection NO-GO trials first surpassed this 95% threshold (*Figure 3D*).

Statistical testing was conducted in Matlab (Mathworks) and the associated Statistics toolbox, and in R. We used the Wilcoxon rank sum test and, for the data set in *Figure 6C,a* generalised linear mixed-effects model. In this model fit, performance on a session was the response variable, task type was the predictor variable, and mouse identity was a random effect: in Wilkinson notation, performance ~1 + tasktype + (1|mouse).

## Human experiments

Human experiments were conducted and underwent ethical review at the University of Sussex. In total, 59 participants were recruited. All gave informed consent. In the human counterpart of the experimental design, the basic GO and NO-GO stimulus waveforms described above (*Figure 1D*) were identical to the mouse version. Further waveforms were added in order to aid and test learning, as described in Results. Stimuli were loaded to the RP2.1 signal processor and delivered via a piezoelectric wafer identical to that used for whisker stimulation, but with a plastic plate glued on (polyethylene terephthalate; 20 × 10 × 1 mm). The wafer stimulator assembly was supported by a platform incorporating a cushioned armrest. Participants were asked to place one fingertip lightly on the plate's surface (*Figure 1C*). The wafer was placed horizontally and vibrations were vertical. A small box containing a button was placed on the same table as the platform, in a position allowing participants to comfortably press the button with their free hand whenever they thought the target stimulus was present. GO and NO-GO stimulus trials were randomly interleaved.

Training was conducted with no explicit instruction as to the identity of the target stimulus. Instead, participants were asked to press the button whenever they identified a stimulus that felt familiar, more frequent or 'special' than others. Therefore, as for the mice, human participants had to work out by themselves which stimulus constituted the target based on the information implicitly available: for humans, this included the number of times the target sequence reoccurred (which was higher than for other stimuli presented) and its similarities and differences with other stimuli. To parallel more closely the experimental design used with mice, feedback upon correct trials, provided in the form of a 'Correct' sign appearing on a computer screen, was given to a subset of participants. We compared performance with and without feedback: performance was no higher for the participants trained with feedback, so results were pooled together (p=0.99, Wilcoxon rank sum test, n = 15 participants without feedback and n = 44 with feedback).

Data used to prepare the figures are provided with this article as linked Source Data files. Additional raw data files will be provided by the corresponding authors on request (m.bale@sussex.ac.uk, m.maravall@sussex.ac.uk).

## Acknowledgements

This work was supported by the Spanish Ministry of Science and Innovation (grant number BFU2011-23049, co-funded by the European Regional Development Fund; Subprograma Ayudas FPI-MICINN, BES-2012–052293), the Medical Research Council (grant number MR/P006639/1), the Valencia Regional Government (ACOMP2010/199 and PROMETEO/2011/086), and the University of Sussex internal research development fund. The authors declare no competing financial interests.

We thank Karel Svoboda for sharing designs, advice and equipment, Leopoldo Petreanu for sharing designs, Rasmus Petersen for sharing code for behaviour control and for comments on an earlier version of the manuscript, and Elena Giusto for technical help and for the drawings in *Figure 1*.

## Additional information

### Funding

| Funder | Grant reference number | Author |
| --- | --- | --- |
| Ministerio de Economía y Competitividad | BFU2011-23049 BES-2012-052293 | Miguel Maravall |
| Generalitat Valenciana | ACOMP2010/199 PROMETEO/2011/086 | Miguel Maravall |
| Medical Research Council | MR/P006639/1 | Miguel Maravall |
| University of Sussex | | Malamati Bitzidou Miguel Maravall |

The funders had no role in study design, data collection and interpretation, or the decision to submit the work for publication.

### Author contributions

MRB, Conceptualization, Data curation, Software, Formal analysis, Validation, Investigation, Visualization, Methodology, Writing—original draft, Writing—review and editing; MB, Software, Formal analysis, Investigation, Methodology, Writing—review and editing; AP, LSB, Investigation, Methodology, Writing—review and editing; LK, STA, CDS, Investigation, Writing—review and editing; MM, Conceptualization, Formal analysis, Supervision, Funding acquisition, Validation, Investigation, Writing—original draft, Project administration, Writing—review and editing

### Author ORCIDs

Michael R Bale, http://orcid.org/0000-0002-5325-1992
Malamati Bitzidou, http://orcid.org/0000-0002-3726-9543
Leonie S Brebner, http://orcid.org/0000-0002-4540-2826
Caitlin D Stevenson, http://orcid.org/0000-0003-4444-6402
Miguel Maravall, http://orcid.org/0000-0002-8869-7206

### Ethics

Human subjects: Human experiments were conducted and underwent ethical review at the University of Sussex. Experiments were approved through the review process in the School of Life Sciences and were given approval identifiers ER/LK250/1, ER/CS502/1, ER/SA533/2. All participants gave informed consent. Participants were provided with an information sheet stating the possibility that the research could be published.

Animal experimentation: All procedures were carried out in accordance with institutional, national (Spain and United Kingdom) and international (European Union directive 2010/63/EU) regulations for the care and use of animals in research. All procedures received prior approval by the relevant institutional ethical committee (Instituto de Neurociencias, CSIC; University of Sussex AWERB) and were covered by the appropriate personal and project licences.

## Additional files

### Supplementary files

• Source code 1. This zip archive includes files used for generating, plotting and loading stimuli and for controlling the behavioural task. Figure1_sequence_generation.m contains Matlab script for generating GO and NO-GO sequences. sequence.m is the main Matlab script for interfacing with TDT ActiveX to control the behaviour: this loads parameters and reads task parameters at the end of a trial. GUI_sequence_v7.fig and GUI_sequence_v7.m are graphical user interface files for monitoring and changing parameters. plot_sequence_v2_3 .m is the Matlab script for plotting task results during the session. LoadRP2.m contains Matlab/ActiveX script to connect to the TDT RP2.1. Finally, TDT_circuit_file.txt contains the TDT ActiveX circuit for controlling the task, exported to C language.

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
