## [Decision Letter]

Thank you for submitting your article "Learning and recognition of tactile temporal sequences by mice and humans" for consideration by *eLife*. Your article has been reviewed by two peer reviewers, and the evaluation has been overseen by a Reviewing Editor and Andrew King as the Senior Editor. The following individual involved in review of your submission has agreed to reveal his identity: Mathew E. Diamond (Reviewer #1).

The reviewers have discussed the reviews with one another and the Reviewing Editor has drafted this decision to help you prepare a revised submission.

There are three major issues in your study that need to be considered in your resubmitted paper:

The first issue is what actually the mice used for their psychophysical reports. It is not clear whether they detected transitions between steps or duration steps or perhaps both as suggested by the reviewers. This needs to be clarified in your submission.

Another important issue is whether the task design is ethological for mice. This is critical for discussing in terms of what the whisker system is really processing. This would require more analysis in your behavioral data.

I am also concerned about whether a comparison study can be made between mice and humans. Of course, it is interesting that humans can perform like rodents and vice versa in the same task. I think your study needs to be more focused on mice performance and as supplementary information present the human performance. Human performance could be used then for the Discussion section.

*Reviewer #1:*

Learning and recognition of tactile temporal sequences by mice and humans by Bale et al. is a fascinating behavioral study about what properties of a vibration mice can feel using their whiskers in the receptive mode, as compared to what human subjects can feel using their fingertips.

The paradigm was very clever and could offer an apparatus and training manual for a large number of labs interested in temporal integration.

The principal stimulus discrimination was a noisy vibration multiplied by an 8-interval step function. One form of the 8-interval function was a GO cue (mouse rewarded for licking) and another form of the 8-interval function was a NO-GO cue: the mouse was allowed to proceed quickly to the next trial if it correctly withheld its licking (correct rejection) but had to undergo an inter-trial delay if it mistakenly licked (false alarm). As expected, human subjects could perform this discrimination well, with over 80% correct. Humans did not seem to detect specific noise signatures, as they were equally as good with frozen and non-frozen sequences.

Unexpectedly (or expectedly, depending on the reader's thinking before starting the manuscript), mice could select between the go and no-go very nicely. The investigators included good controls, like removing whiskers and thus causing many miss trials. Of course, Go and No-go had overall same energy (though interestingly they later do an experiment showing that mice can detect overall energy level). Also interesting that performance generalized across movement orientation (motor rotated) and I think the authors should use this observation of generalization to say something about what must be detected by the mice.

Where mice fell down in performance was when amplitude steps were concatenated in a way that go and no-go stimuli had the same sequence of transitions, but took continuous amplitude levels for different time lengths. This makes the authors believe that mice were focused on detecting transitions between steps but were not sensitive to durations of steps. That is fine, but two points of caution are in order: First, the mice were still above chance even with step durations mixed. The authors see the glass half-empty as it were ("performed poorly"), because they report the drop-in performance without emphasizing that mice still got it… at least they were above chance. The second point is that it's always risky to draw too strong a conclusion from a negative result. People know that rodents (rats, at least) are good at measuring stimulus duration. So they do not measure only the state (amplitude) and transition, but also the elapsed time. The mechanism exists in the brain. Could they not use their time-measurement capacity in the present paradigm? Apparently they had some deficit. But it's always possible that a different training mechanism – one that began with training mice to distinguish two durations of the same noise amplitude, before concatenating – could lead them to perform well.

In sum, there's much more to do, especially – as the authors point out – in finding neuronal bases for pattern recognition. The current manuscript does a solid job of laying down a very rich new paradigm for rodent temporal integration studies.

*Reviewer #2:*

Bale et al. present a set of observations comparing human and mouse touch psychophysics in a go/no-go paradigm. The methods and experimental designs are clever (like comparing frozen vs. unfrozen sequences, for example) and clearly presented. Very briefly, the authors show that humans perform better than mice on a task, which requires comparing a temporal sequence of amplitudes of a tactile stimulus. The paper is well written and the authors nicely discuss/speculate how their observations align with findings in neurophysiology, but it's less clear whether the paper provided sufficient novelty to justify publication in *eLife*.

Is the task-design really ethological for mice? Based on the poor performance of mice, as compared to humans, the authors go on to speculate which kind of information cortical circuits in the mouse can extract from sensory stimuli. I think that this speculation is somewhat unjustified. The whisker system is made for active touch, not passive vibration, and it may well be that the whisker system is optimized to extract information during active touch, and thus performs poorly in a passive setting. For example, we know that both human S1 (Simões-Franklin et al., Hum. Brain Mapp. 2011) and barrel cortex (Crochet et al. Neuron 2011) and even first-order neurons in the trigeminal ganglion (Szwed et al. Neuron 2003) respond differentially to active and passive touch. The amplitudes and durations chosen for this task were arbitrarily set by the investigators and may to correspond to the natural scene statistics, which the whisker system is specialized to sample in nature. This is a caveat, which should be more clearly discussed by the authors.

In Figure 4, the authors show that – for mice – the noise amplitude sequences [3 1 4 2 3 1 4 2] v [3 4 2 1 2 4 3 1] are more distinguishable than the sequences [4 1 4 1 1 1 4 1] v [4 1 1 1 4 1 4 1]. They conclude that:

"This suggested that mice either did not detect the simpler, binary stimulus modulation epochs or did not recognise their differential duration. To distinguish between these possibilities, we tested performance on probe sessions with two variants of the binary NO-GO sequence."

The authors then go on to conclude:

"The overall conclusion of the binary sequence experiments is that mice could detect "large" epochs and recognise their number, and use the presence of relative modulations in noise amplitude as cues, but could not as readily use the duration of each modulation epoch." The authors should discuss another view, namely that the mice are paying attention to the derivative of the sequence. The derivative of the sequences [4 1 4 1 1 1 4 1] and [4 1 1 1 4 1 4 1] are [-3 +3 -3 0 0 +3 -3] and [-3 0 0 +3 -3 +3 -3]. Thus, they both have the pattern [-3 +3 -3 +3 -3]. This view is related to the alternative mentioned by the authors, but discussing it more explicitly would improve the manuscript. Perhaps the authors should discuss this in terms of the whisker system, which may be more specialized for detecting changes rather than the constant "energy" of a vibrational noise stimulus?

The authors do not explicitly mention that in the comparison [3 1 4 2 3 1 4 2] v [3 4 2 1 2 4 3 1], the first sequence repeats itself. Does this periodicity explain the higher performance? Is it easier to detect a periodic sequence?

Sometimes, the authors present results of generalized linear mixed-effects models, e.g.:

"In both of these variants animals performed better than in the original binary design (Figure 4 < 10-9; n = 7 mice and n = 161 sessions; generalised linear mixed effects model)." In such cases, the authors should also present the fitted betas, so that the reader can know the magnitudes of the effects. In the above case, for example, we are told that the performance is significantly better, but we have no way of knowing how much better it is. It would also benefit the reader to know exactly how the GLME was constructed and grouped – by animal, by session, etc.? This could be added as a small section to the Materials and methods, which lists all the fitted GLMEs in Wilkinson notation, for example.

In the human experiments, the participants were allowed to choose for themselves which sequence they preferred to respond to, while the mice were all forced to respond to the same sequence. Does this introduce a bias, which makes the humans appear much better than the mice?

The authors have designed their task as a go/no-go task, which has some inherent problems. For example (as the authors also mention in their Materials and methods section), it is impossible to know if a no-go response represents a 'correct rejection' or simply lack of motivation to respond in that given trial. Thus, in general, there are theoretical reasons to prefer forced-choice designs to go/no-go designs (as reviewed e.g. in Churchland and Carandini, Nat.Neurosci. 2013). Perhaps the authors could discuss if their design could be re-framed as a forced-choice design?

---

## [Author Response]

*There are three major issues in your study that need to be considered in your resubmitted paper:*

*The first issue is what actually the mice used for their psychophysical reports. It is not clear whether they detected transitions between steps or duration steps or perhaps both as suggested by the reviewers. This needs to be clarified in your submission.*

We have made significant changes to address this point. We have added a new analysis of lick response latencies which demonstrates the ability of mice to respond selectively, early in the stimulus sequence (almost as soon as it was possible to detect whether the sequence was GO or NO-GO by virtue of the first transition between steps) (subsection “Achieving sequence recognition by mice” fourth and sixth paragraphs, subsection “Task control and analysis”, fourth paragraph, new Figure 3 and Figure 4). Animals were therefore effective at detecting steps and the transitions between them. However, under our conditions, they were clearly not as successful at measuring the duration of steps as they were at detecting transitions, and we now state this more clearly (subsection “Binary sequence discrimination”, fifth paragraph), including in the Abstract (“Mouse sequence recognition depended on detecting transitions in noise amplitude; animals could base their decision on the earliest information available”). We have rewritten several parts of the manuscript (detailed below in our specific responses to reviewers) to put across a clearer message as to the strategies used by mice in the task.

On the other hand, we have also rewritten some passages to clarify that the fact that mice did not adopt other potential strategies under our conditions does not imply that those strategies were necessarily unavailable to them (e.g. Discussion, fifth paragraph). This important point was raised by reviewer 1 and we detail our response below.

*Another important issue is whether the task design is ethological for mice. This is critical for discussing in terms of what the whisker system is really processing. This would require more analysis in your behavioral data.*

We discuss this issue in detail in our response to reviewer 2. We recognise that both in mice and in humans, our task design sought to isolate capacities for discriminating temporally patterned sequences, rather than to exactly mirror a natural situation faced by the animal. Our approach follows extensive precedent in both neuroscience and comparative ethology, as mentioned in the manuscript’s Discussion (Discussion, fourth paragraph). Importantly, our data indicate that *mice were able to learn and recognise sequences at a level comparable to humans*, so that the task design had similar validity in both species, suggesting that rodents have general capacities for extracting cues for sequence identity. Our new behavioural analysis of lick latencies throws further light onto what is being processed by the animal, by showing that mice can reach their decision on the identity of the sequence as soon as it is feasible to do so (subsection “Achieving sequence recognition by mice”, sixth paragraph, subsection “Potential cues for sequence recognition”, last paragraph and new Figure 3 and Figure 4). We also mention possible consequences for the experimental design that arise from the actively sensing nature of the whisker system (subsection “Stimulus delivery and design”, last paragraph). As mentioned, all of these points are further detailed in our response to reviewer 2.

*I am also concerned about whether a comparison study can be made between mice and humans. Of course, it is interesting that humans can perform like rodents and vice versa in the same task. I think your study needs to be more focused on mice performance and as supplementary information present the human performance. Human performance could be used then for the Discussion section.*

We have thought hard about this issue. The paper does provide a side-by-side comparison of human and animal performance. Performance levels, as detailed below, overlap substantially. It is true that the two species do not completely mirror each other’s behaviour, despite the overlap in performance: the strategies used in the task differ at least in part. However, we feel that our results on human perception provide important context for the animal findings. They enable a comparison whereby a numerical result for animal performance can be scaled against the human (and we now explicitly facilitate this, e.g. in Figure 5). Ultimately, this comparative approach facilitates calibration of the animal data and allows the reader to judge whether the task can be generalized, and whether animals and humans follow similar strategies (Brunton et al., Science 2012; Fassihi et al., PNAS 2014). For this reason, we feel that providing the human results is valuable to our account. In any case, because we have now enriched our analysis of mouse performance (new panels in Figure 3–Figure 4, together with corresponding new paragraphs in the text), the manuscript now places more emphasis on mouse behaviour. We believe the effect of this has been to implement the advice of the Reviewing Editor, in that the human data now read mainly as being of comparative value.

*Reviewer #1:*

*[…] The principal stimulus discrimination was a noisy vibration multiplied by an 8-interval step function. One form of the 8-interval function was a GO cue (mouse rewarded for licking) and another form of the 8-interval function was a NO-GO cue: the mouse was allowed to proceed quickly to the next trial if it correctly withheld its licking (correct rejection) but had to undergo an inter-trial delay if it mistakenly licked (false alarm). As expected, human subjects could perform this discrimination well, with over 80% correct. Humans did not seem to detect specific noise signatures, as they were equally as good with frozen and non-frozen sequences.*

The reviewer is correct about this. We note that there is an interesting footnote to the result on frozen and non-frozen sequences. A new non-frozen waveform was generated for every trial, representing a new random sample of stimulus values. Because of random variation, the waveform for each such sample had a different empirical standard deviation: for example, some waveforms (i.e. some trials) included prominent stimulus “events” or “landmarks”. To properly calibrate our comparison between frozen and non-frozen sequences, we excluded all non-frozen trials with a deviant SD, i.e. we included only non-frozen trials where the SD was comparable to that for the frozen waveform. However, if one does consider trials with high SD, these showed a higher hit rate than the frozen ones, suggesting that humans *can* detect and respond to specific events or signatures if available. We briefly mention this point at the end of the Results section but have chosen not to dwell on it further, in order not to complicate our account of the main results.

*Unexpectedly (or expectedly, depending on the reader's thinking before starting the manuscript), mice could select between the go and no-go very nicely. The investigators included good controls, like removing whiskers and thus causing many miss trials. Of course, Go and No-go had overall same energy (though interestingly they later do an experiment showing that mice can detect overall energy level). Also interesting that performance generalized across movement orientation (motor rotated) and I think the authors should use this observation of generalization to say something about what must be detected by the mice.*

We thank the reviewer for these positive comments. Although these observations appear at a relatively early stage in the Results, where it is hard to be more specific about what mice were detecting, we have added some more explicit passages:

“[t]he new configuration involved a different array of forces and moments acting upon a different set of whisker follicles.”

“[t]he animal must be focusing on aspects of the pattern of fluctuations over time, rather than on movements of a specific whisker or in a specific direction.”

*Where mice fell down in performance was when amplitude steps were concatenated in a way that go and no-go stimuli had the same sequence of transitions, but took continuous amplitude levels for different time lengths. This makes the authors believe that mice were focused on detecting transitions between steps but were not sensitive to durations of steps. That is fine, but two points of caution are in order: First, the mice were still above chance even with step durations mixed. The authors see the glass half-empty as it were ("performed poorly"), because they report the drop-in performance without emphasizing that mice still got it… at least they were above chance.*

We take the reviewer’s point and have rewritten this with more nuance, including removal of “performed poorly”:

“We found that mice did not perform well at distinguishing [4 1 4 1 1 1 4 1] from [4 1 1 1 4 1 4 1] (…) Performance was above chance but never reached 75% (Figure 6; n = 3), in contrast to the original variant.”

*The second point is that it's always risky to draw too strong a conclusion from a negative result. People know that rodents (rats, at least) are good at measuring stimulus duration. So they do not measure only the state (amplitude) and transition, but also the elapsed time. The mechanism exists in the brain. Could they not use their time-measurement capacity in the present paradigm? Apparently they had some deficit. But it's always possible that a different training mechanism – one that began with training mice to distinguish two durations of the same noise amplitude, before concatenating – could lead them to perform well.*

Again, we recognise the reviewer’s point. We now state that the fact these cues were not used by mice was specific to the present paradigm. We have also rewritten and inserted qualifiers in all relevant passages, to remove any implication that mice were necessarily unable to extract cues or mechanisms not used in this task: “in this paradigm mice could have had trouble…”, “under these conditions humans appeared to readily access more cues for sequence discrimination than mice,”. The Discussion now states:

“[w]e cannot rule out that mice have access to the same cues as humans but that only some cues were engaged by our design. With these caveats, our results suggest that mice relied primarily on particular transitions in stimulus amplitude (…)”

“Further testing of mouse capacities for using additional cues, from epoch duration up to more abstract sequencing rules, is needed”.

*In sum, there's much more to do, especially – as the authors point out – in finding neuronal bases for pattern recognition. The current manuscript does a solid job of laying down a very rich new paradigm for rodent temporal integration studies.*

*Reviewer #2:*

*Bale et al. present a set of observations comparing human and mouse touch psychophysics in a go/no-go paradigm. The methods and experimental designs are clever (like comparing frozen vs. unfrozen sequences, for example) and clearly presented. Very briefly, the authors show that humans perform better than mice on a task, which requires comparing a temporal sequence of amplitudes of a tactile stimulus. The paper is well written and the authors nicely discuss/speculate how their observations align with findings in neurophysiology, but it's less clear whether the paper provided sufficient novelty to justify publication in eLife.*

We thank the reviewer for their comments. We must however disagree that humans always perform better than mice or that mice perform poorly. Comparing performance for mice and humans in the main version of the task (original Figure 2 and Figure 3, current Figure 4 and Figure 5) shows that mice often perform better than humans. Median performance for mice was 74% and the best individual mice reached 84%. Median performance for humans was 70%. Of course, rigorous numerical comparison is complicated by the fact that mice underwent training for longer than humans, and human performance could have improved with greater practice. However, there is no doubt that mice could learn the task, just as the successful humans did. This is now more explicitly indicated in the Results:

“Despite unavoidable differences in training procedure, mice and humans reached a similar median level of performance (mice: 74%; humans: 70%), and the range of performance across individuals also overlapped substantially between the two species (Figure 5).”

And in the Discussion:

“There was substantial overlap between the performance levels of mice and humans (Figure 5).”

Thus, individual mice performed better than many individual humans, and there was substantial overlap in performance between mice and humans. What does appear distinct between humans and mice, as explored in Figure 6, is the nature of the cues (information) used to solve the task. While humans often seemed to use a “holistic” or “global” strategy that pays attention to the overall patterning of the sequence (Figure 6, subsection “Binary sequence discrimination”, last paragraph, subsection “Fixed landmarks versus amplitude modulation”, second paragraph), mice appeared to rely principally on detecting particular changes or transitions in stimulus amplitude between parts of the sequence. We have significantly added to our behavioural analysis in order to strengthen and clarify this point. Notably, we present a new analysis of lick response latencies in the main task (“discriminative lick latency”; new panels in Figure 3 and Figure 4, associated subsection “Achieving sequence recognition by mice”, fourth paragraph and sixth paragraphs; subsection “Potential cues for sequence recognition”, last paragraph; subsection “Task control and analysis”, fourth paragraph). This shows that animals trained on the task responded differentially to GO versus NO-GO – i.e. reached their decision on sequence identity – within 200 ms in the fastest cases, i.e. after having felt just one transition in the amplitude envelope; and before the end of the sequence in all cases. The implication is that mice sought to detect differences between GO and NO-GO stimuli without integrating the entire sequence or waiting until the end.

*Is the task-design really ethological for mice? Based on the poor performance of mice, as compared to humans, the authors go on to speculate which kind of information cortical circuits in the mouse can extract from sensory stimuli. I think that this speculation is somewhat unjustified.*

As noted in the previous point, we must respectfully disagree with the premise that mice had “poor performance” on the main task as compared to humans: the data do not support the conclusion that mice were unsuited to the task. Rather, mice could adapt their capacity for sequence recognition to the conditions of our experimental design, and successfully learn the task at a level of performance comparable to humans.

We would also disagree that our account of the information used by mice is simply speculation. Our account is based on quantitative comparisons of performance across stimuli that are easier or harder to distinguish, and (in this revision) on measurements of lick latency that demonstrate when animals have reached a decision on sequence identity. These establish that animals can base their decision on information available within 200 ms from the beginning of the sequence. This and further analyses imply that mice must be deciding based on detecting modulations (or transitions) in stimulus amplitude.

*The whisker system is made for active touch, not passive vibration, and it may well be that the whisker system is optimized to extract information during active touch, and thus performs poorly in a passive setting. For example, we know that both human S1 (Simões-Franklin et al., Hum. Brain Mapp. 2011) and barrel cortex (Crochet et al. Neuron 2011) and even first-order neurons in the trigeminal ganglion (Szwed et al. Neuron 2003) respond differentially to active and passive touch. The amplitudes and durations chosen for this task were arbitrarily set by the investigators and may to correspond to the natural scene statistics, which the whisker system is specialized to sample in nature. This is a caveat, which should be more clearly discussed by the authors.*

We agree with the reviewer that whisker-mediated exploration seems to occur primarily through active touch, i.e. through active whisker motion. Whiskers actively scan the environment for objects and this exploration is under exquisite behavioural control. For example, rodents can concentrate their whisking on a focal point which may differ on either side of the snout, and can quickly regulate the force and spread of whisker movement upon contacting a surface. Importantly, this behavioural control also extends to minimising whisker motion when an animal expects to receive externally driven tactile stimulation, such as in the context of a laboratory sensory discrimination task (Miyashita and Feldman, Cereb Cortex 2012; Sachidhanandam et al., Nature Neurosci 2013; Fassihi et al., PNAS 2014). Thus, when trained to detect or discriminate between whisker stimuli received passively, mice and rats can learn to “prepare for” receiving the stimulus, even to the point of keeping the snout and whiskers almost immobile. As demonstrated by the above-cited studies and other prominent work by multiple labs (e.g. Nicolelis or Schwarz groups), both mice and rats can successfully process passive stimuli. Performance levels achieved by mice in our study are comparable to humans (as argued earlier) and also to animals in these earlier studies.

We would suggest that, similarly, a lab situation where a human participant awaits delivery of a stimulus to the fingertip is not representative of natural patterns of human tactile exploration, in that these tend to involve active scanning of textures and objects to collect information. And as the reviewer mentions, neurons in human S1 also respond differentially to active and passive touch. Yet introspection and experimental results tell us that humans can adapt their tactile discrimination strategies to passive stimulation if needed. We would argue that, based on the literature as well as our own results, rodents do likewise.

We agree with the reviewer’s point that a caveat about the passive nature of these stimuli, given that the system appears to use mostly active touch, is in order. We have added the following to the Discussion: “In both rodents and humans, tactile sensation commonly involves active exploration, whereby the animal generates sensor motion in order to feel the object. Our design departed from active sensing in that mice and humans were trained to receive and recognise sequences delivered by a stimulator.” This is followed by further discussion of the relevance of using passive stimulation.

We also agree that the parameters and durations chosen for the task were set by us and may not correspond to natural scene statistics. In our view, that mice learned the task anyway is evidence for the animals’ ability to adapt their capacities to the situation.

Finally, and also relevant to the reviewer’s point, aspects of our stimulus design were set up to try to provide robustness against possible interference between passive stimulation and the animals’ natural tendency to whisk. At the reviewer and Reviewing Editor’s suggestion, we have added a new paragraph to the Materials and methods, discussing these aspects of the stimulus; for brevity, we only cite in part here):

“We considered other approaches to sequence design before opting for continuous noise stimulation. For example, a sequence can be constructed as a series of discrete whisker pulse stimuli separated by pseudorandom intervals [Pitas et al., 2017]. We preferred continuous stimulation (…) any active whisks generated by the animal could interfere with its judgment of the duration of intervals. This potential conflict, inherent to the use of intact whiskers as active sensors, is reduced in the case of continuous stimulation where the animal is being asked to judge the amplitude of “syllables” lasting 100 ms.”

*In Figure 4, the authors show that – for mice – the noise amplitude sequences [3 1 4 2 3 1 4 2] v [3 4 2 1 2 4 3 1] are more distinguishable than the sequences [4 1 4 1 1 1 4 1] v [4 1 1 1 4 1 4 1]. They conclude that:*

*"This suggested that mice either did not detect the simpler, binary stimulus modulation epochs or did not recognise their differential duration. To distinguish between these possibilities, we tested performance on probe sessions with two variants of the binary NO-GO sequence."*

*The authors then go on to conclude:*

*"The overall conclusion of the binary sequence experiments is that mice could detect "large" epochs and recognise their number, and use the presence of relative modulations in noise amplitude as cues, but could not as readily use the duration of each modulation epoch." The authors should discuss another view, namely that the mice are paying attention to the derivative of the sequence. The derivative of the sequences [4 1 4 1 1 1 4 1] and [4 1 1 1 4 1 4 1] are [-3 +3 -3 0 0 +3 -3] and [-3 0 0 +3 -3 +3 -3]. Thus, they both have the pattern [-3 +3 -3 +3 -3]. This view is related to the alternative mentioned by the authors, but discussing it more explicitly would improve the manuscript. Perhaps the authors should discuss this in terms of the whisker system, which may be more specialized for detecting changes rather than the constant "energy" of a vibrational noise stimulus?*

This is a good point. We agree that what the reviewer terms “detecting changes” seems to correspond to what we term “detecting transitions”. We also agree that this is likely to be a cue used by mice. We now include the point about the pattern of changes or transitions being [-3 +3 -3 +3 -3] in both of the GO and NO-GO stimuli mentioned by the reviewer:

“Note that the pattern design was such that the actual sequence of transitions in amplitude was identical between GO and NO-GO: transitions followed the order [-3 +3 -3 +3 -3].”

We also mention the transition pattern [-3 +3] in the simpler binary variant:

“There were now just two transitions in amplitude, in the order [-3 +3].”

Finally, we mention the actual word “changes” rather than “transitions” in amplitude at several points in the manuscript (Results and Discussion).

*The authors do not explicitly mention that in the comparison [3 1 4 2 3 1 4 2] v [3 4 2 1 2 4 3 1], the first sequence repeats itself. Does this periodicity explain the higher performance? Is it easier to detect a periodic sequence?*

We thank the reviewer for pointing this out. The “discriminative lick latency” analysis described above shows that mice were able to reach a decision on the identity of the sequence within 200 ms from stimulus onset. This implies that, for mice, later repetition in the sequence was unlikely relevant, at least once the GO sequence was learned (it could be that mice do use periodicity as a cue during learning and then, once they are good at performing, switch to the fastest possible strategy). For humans, given that the global structure of the sequence did seem to be an important cue, periodicity (felt as stimulus “rhythm” or “number of beats”) may have played a role. We state this in the Discussion:

“Participants receiving the GO sequence often reported feeling a vibration consisting of a rhythmic series of buzzes, or counting “beats” in the stimulus. This denoted an ability to detect stimulus periodicity or repeats.”

*Sometimes, the authors present results of generalized linear mixed-effects models, e.g.:*

*"In both of these variants animals performed better than in the original binary design (Figure 4 < 10-9; n = 7 mice and n = 161 sessions; generalised linear mixed effects model)." In such cases, the authors should also present the fitted betas, so that the reader can know the magnitudes of the effects. In the above case, for example, we are told that the performance is significantly better, but we have no way of knowing how much better it is. It would also benefit the reader to know exactly how the GLME was constructed and grouped – by animal, by session, etc.? This could be added as a small section to the Materials and methods, which lists all the fitted GLMEs in Wilkinson notation, for example.*

We ran just one generalized linear mixed-effects model, to evaluate the dependence on task type of the data in Figure 4 (now 6C). As requested, we now present the fitted coefficient for task type. We present it as a t-statistic (i.e. the z-score value): “t-statistic for the regression on task type = 6.82”. We also provide a more precise p value here, and have taken the opportunity to correct the number of animals used in these experiments, which was misquoted in the earlier version: “n = 10 mice and n = 161 sessions”.

We have also added the following information on the GLME model in Materials and methods, as requested:

“We used (…) a generalised linear mixed-effects model. In this model fit, performance on a session was the response variable, task type was the predictor variable, and mouse identity was a random effect: in Wilkinson notation, performance ~ 1 + tasktype + (1|mouse).”

*In the human experiments, the participants were allowed to choose for themselves which sequence they preferred to respond to, while the mice were all forced to respond to the same sequence. Does this introduce a bias, which makes the humans appear much better than the mice?*

This is a misunderstanding – we apologise for not providing sufficient clarity. Actually, the target GO sequence for the human participants was fixed by us in advance, with no intervention by the participants, exactly as for the mice. The sequence was in fact identical to that used for mice. In training, human participants were just asked to determine the correct sequence to the best of their ability, based on their growing familiarity with the stimulus set – again, similar to the mice. They were not explicitly instructed to identify a sequence with certain characteristics, but had to figure out for themselves which might be the target: specifically, they were asked only to identify the sequence that felt “special” or “familiar”. This implicitly directed them to consider the number of times the sequence reoccurred (higher than for other stimuli), and its similarities and differences with other stimuli. We now include this information more clearly and in greater detail in the relevant Materials and methods paragraph, which has been rewritten:

“Training was conducted with no explicit instruction as to the identity of the target stimulus. Instead, participants were asked to press the button whenever they identified a stimulus that felt familiar, more frequent or “special” than others. Therefore, as for the mice, human participants had to work out by themselves which stimulus constituted the target based on the information implicitly available: for humans, this included the number of times the target sequence reoccurred (which was higher than for other stimuli presented) and its similarities and differences with other stimuli.”

We have also included this point in the relevant Results paragraph for greater clarity: “the GO target sequence – identical to that presented to mice – was interleaved with a series of non-target stimuli”.

*The authors have designed their task as a go/no-go task, which has some inherent problems. For example (as the authors also mention in their Materials and methods section), it is impossible to know if a no-go response represents a 'correct rejection' or simply lack of motivation to respond in that given trial. Thus, in general, there are theoretical reasons to prefer forced-choice designs to go/no-go designs (as reviewed e.g. in Churchland and Carandini, Nat.Neurosci. 2013). Perhaps the authors could discuss if their design could be re-framed as a forced-choice design?*

This is an important point; we thank the reviewer for raising it. We have indeed tested this. We have begun to develop a variant of the task where mice learn to discriminate auditory rather than tactile sequences (demonstrating generalisation across modalities) and, moreover, do so within a two-alternative forced choice design as suggested by the reviewer. In this protocol, freely moving mice learn to wait at a central nose poke at the beginning of a trial and to move to either a left or a right nose poke depending on the sequence being presented. We now mention this in the Discussion:

“[m]ice robustly learn a simpler version of the task where the GO sequence is structured as XABX and the NO-GO sequence as XBAX (with X, B and A denoting different waveforms that, concatenated, make up the sequence; Bale, Bitzidou, Giusto and Maravall, data not shown). […] Mice also learn to distinguish XABX versus XBAX in an auditory two-alternative forced choice design for freely moving animals (Saska, Giusto, Bale, Bitzidou and Maravall, data not shown), suggesting that this form of sequence learning generalises across sensory modality and protocol.”